

# Brief communication: Analysis of organic matter in surface snow by PTR-MS – implications for dry deposition dynamics in the Alps

Dušan Materić[1], Elke Ludewig[2], Kangming Xu[1], Thomas Röckmann[1], Rupert Holzinger[1]

[1]Institute for Marine and Atmospheric Research Utrecht, Utrecht University, Princetonplein 5, 3584CC
Utrecht, The Netherlands
[2]ZAMG - Zentralanstalt für Meteorologie und Geodynamik, Sonnblick Observatory, A-5020 Salzburg,
Freisaalweg 16, AUSTRIA

*Correspondence to*: Dušan Materić (dusan.materic@gmail.com)

**Abstract.** The exchange of organic matter (OM) between the atmosphere and snow is poorly
understood due to the complex nature of OM and the convoluted processes of deposition, re-
volatilisation, chemical, and biological processing. OM that is finally retained in glaciers potentially
holds a valuable historical record of past atmospheric conditions; however, our understanding of the
processes involved is insufficient to translate the measurements into an interpretation of the past
atmosphere. This study examines the dynamic processes of post-precipitation OM change at the alpine
snow surface with the goal to interpret the processes involved in surface snow OM.

## 1 Introduction

Organic matter (OM) in the cryosphere originates from different sources (e.g. oxidation products of
anthropogenic, biogenic, and biomass burning VOCs), is transported from short and long distances and
is deposited via dry or wet deposition (Antony et al., 2014). From the moment of the emission, OM
undergoes atmospheric chemistry processes, which profoundly alter the chemical composition of OM,
resulting in numerous chemical species that are finally deposited on the snow/ice surface (Legrand et
al., 2013; Müller-Tautges et al., 2016). Fingerprints of OM stored in the snow and ice therefore
potentially hold a rich historical record of atmospheric chemistry processes and the transport pathways
in the atmosphere (Fu et al., 2016; Giorio et al., 2018; Grannas et al., 2006; Pokhrel et al., 2016).
The vast diversity of OM, which is found in snow and ice samples, is impossible to characterise by one
single method. The most used methods so far in snow/ice OM research are based on gas
chromatography (GC) and liquid chromatography mass spectrometry (LC-MS) (Giorio et al., 2018;
Gröllert et al., 1997). Novel high-resolution mass-spectrometry-based analytical methods, such as
Fourier-transform ion cyclotron resonance mass spectrometry (FT-ICR-MS), Orbitrap mass
spectrometry, and Thermal Desorption – Proton Transfer Reaction – Mass Spectrometry (TD-PTR-MS),
have recently been developed and can be used to characterise OM in the cryosphere with high mass
resolution (Hawkes et al., 2016; Kujawinski et al., 2002; Marsh et al., 2013; Materić et al., 2017) .





Therefore, numerous new proxies are now potentially available to interpret the rich composition of OM
in the cryosphere.

Reconstructing past atmospheric conditions from measurements of OM in the cryosphere is analytically
challenging because of (1) low concentrations of target organics in the sample, and (2) chemical
changes that (might) happen after OM deposition. A recently developed method using Thermal
Desorption – Proton Transfer Reaction – Mass Spectrometry (TD-PTR-MS) has partly solved the first
issue enabling the detection of low molecular weight OM ranging from 28 to 500 amu (Materić et al.,
2017) . However, chemical changes (e.g. photochemical, biological etc.) and re-emission from the
snow/ice surface still remain challenging to quantify, especially in the context of the diversity of OM
species, both high and low molecular weight OM.

The low molecular weight fraction represents an important part of OM in the cryosphere and the group
includes volatile and semi-volatile organic compounds (VOC, sVOC) that deposit directly from the gas
phase or as part of secondary organic aerosols (SOA). Low molecular weight OM has been extensively
studied in atmospheric context by real-time or off-line PTR-MS techniques (Gkatzelis et al., 2018;
Holzinger et al., 2010a, 2013; Materić et al., 2015; Timkovsky et al., 2015), however, not so in the
context of deposited (e.g. dissolved) OM in the cryosphere. In this work, we applied a novel TD-PTR-
MS method to measure concentrations of OM present in Alpine snow. The first application of this new
technique is investigation of snow-atmosphere interaction of OM during a dry weather period.

## 2 Material and methods

### 2.1 Sampling site

The snow samples were taken at 3.106 m altitude at Mt. Hoher Sonnblick, Austria, close to the research
station Sonnblick Observatory. The sample site was next to the southern precipitation-measuring
platform, which is about 50 m southeast of the observatory. The sampling location was carefully chosen
to be least affected by potential contamination coming from the observatory. Average temperature of
the site is about 1.1°C in summer and -12.2°C in winter considering the meteorological data being
gathered since 1886.

The sample period spans the days from March 20[th] to April 1[st] of 2017. During this time period the
Sonnblick Observatory experienced an average day length of 12.5 hours, and an average temperature of
-4°C, 78% relative humidity, an average wind speed of 7.3 m s[-1] and pressure of 696 hPa. There was no
significant precipitation observed but these days were mostly foggy in the morning with the exception
of March 27[th] and 28[th] 2017, which were nearly clear-sky days, followed by less cloudy days till April
1[st] 2017. The measured air temperatures (2 m above the surface) on the site were below zero for all the
time, with exception of 3 brief instances where the temperature was recorded 0.1°C for 10 minutes (Fig.
1a). However, hourly temperature averages for these events were also < 0°C. If we use a Positive
Degree Day model (PDD) to assess the melting possibility of those single 10 minutes periods, we
calculated the depth of the melt water 1.4-5.5 μm (using the snow melt factor 2-8 mm °C[-1] day[-1]) (Singh
et al., 2000). Thus, we conclude that no significant melting and runoff happened for the entire sampling
period.



More information on the meteorological conditions can be found in Fig. 1 and Fig. A1.

## 2.2 Sampling

Snow samples were taken from the surface snow (<2 cm) scooping the snow directly into clean 50 mL polypropylene vials. We also took field blanks (ultrapure water) to assure that our blanks were exposed to the same impurities as the snow samples. The samples were stored in a freezer at -20°C until the end of the sampling campaign and then shipped on dry ice to the analysis lab, where they were kept frozen until the analysis.

## 2.3 Analysis

Prior to the analysis the samples (and blanks) were melted and filtered through a 0.2 μm PTFE filter. We loaded 1 mL of each sample into clean 10 mL glass vials that had been prebaked at 250°C overnight. The samples (in triplicates) together with the field blanks, were dehydrated using a low-pressure evaporation/sublimation system and analysed by TD-PTR-MS (PTR-TOF 8000, IONICON
Analytik), following the method described before (Materić et al., 2017) . The thermal desorption procedure was optimized for snow-sample analysis and has the following temperature sequence: (1) 1.5 min incubation at 35 °C, (2) ramp to 250 °C at a rate of 40 °C min$^{-1}$, (3) 5 minutes at 250 °C, and (4) cooling down to <35 °C. The method is fast (<15 min per run), sensitive (e.g. LoD <0.17 ng mL$^{-1}$ for pinonic acid, LoD<0.26 ng mL$^{-1}$ for levoglucosan), requires a small sample size (<2 mL of water), and
it provides reasonably high mass resolution data (>4500, FWHM).
For the data analysis, we used the custom made software package PTRwid for peak integration and identification, and R scripts for statistical analyses (linear regression, fitting etc.) (Holzinger, 2015). We used 3σ of the field blanks for estimating the limit of the detection (LoD), so only ions that are above this value were taken into account for the scientific interpretation (Armbruster and Pry, 2008). We
evaluated the impurities in the field blanks by comparing them with the system blanks (clean vials) and discovered that the average impurity level of a field blank was reasonably low (7.0 ng mL$^{-1}$), which mostly (60 %) originated from the ion m/z 81.035 ($C_5H_4OH^+$). The impurities here might originate from the polypropylene vials we used, however, the levels are much lower compared to the methods used for measuring total and dissolved OM (Giorio et al., 2018). The impurities were taken into account by
means of field blank subtraction and LoD filtering (Materić et al., 2017).
From the mass spectra, identified peaks were integrated over 8 minutes starting when the temperature in the TD system reached 50°C. Extracted peaks were quantified by PTRwid and the concentration was expressed in ng mL$^{-1}$ of sample. We calculated the molar concentration of C, H, O and N for each sample, from which atomic ratios (O/C, H/C, N/C), mean carbon number (nC), and mean carbon
oxidation state (OSC) are calculated as described earlier (Holzinger et al., 2013; Materić et al., 2017). For the elemental composition calculation, we excluded ions m/z <100 as these are dominated by thermal dissociation products of non-volatile high molecular weight compounds. Taking into account these fragments of bigger molecules would substantially alter elemental composition and atomic ratios.





## 3 Results and Discussion

### 3.1 Total ion concentration and simple mass balance model

During our sampling period, the total concentration of organics increases in general over the time that the snow was exposed to the atmosphere (Fig. 2a). The concentration of organics in the snow surface reflects a dynamic balance between two opposing processes that work independently: deposition as source and loss. If we consider just dry deposition (it was a period without precipitation), the retained
(actual) concentration of the organics in the snow can be described as:

$$\frac{dm}{dt} = D - L, \qquad (1)$$

where $m$ is the concentration of organics remaining in the snow, $D$ is the total dry deposition rate and $L$ is the overall loss rate due to re-volatilisation, photochemical reactions, biological processes etc. As our samples generally show an increase in the ion concentrations (Fig. 2 and 3), the loss rate by re-
volatilisation, photochemical reaction and biological decay is lower than the total deposition rate ($D > L$). A negative mass balance, i.e. $D < L$, can happen, for example, in periods of extensive photochemical reactions together with snow exposure to an air mass with a low concentration of OM.
Our total concentration data (as well as many individual ion groups, see below) indicate a relaxation towards a source-sink equilibrium. Mathematically, the simplest model that has these characteristics is a
system with quasi-constant deposition rate $D$ (i.e. changes in deposition are much slower than changes in the loss rate) and a first order loss rate ($L=-km$ in equation 1), which can be integrated to yield

$$m = m_0 e^{-kt} + \frac{D}{k} (1 - e^{-kt}), \qquad (2)$$

where $m_0$ is the initial concentration of $m$, $k$ is the first order loss rate coefficient and $t$ is time. In our experiment, we measured $m$ with a time step $t$ of three days and consider $m_0$ as our measurement of the
fresh snow in the beginning of the analysis period. Eq. 2 can then be fit to the data and the best fit for the total concentration of semi-volatile organic traces ($R^2 > 0.98987$ and rRMSE $< 3.5\%$) was found for $k = 0.31$ day$^{-1}$ and $D = 206$ ng mL$^{-1}$ day$^{-1}$. When the fit is applied to the mass of carbon in the detected organics the best fit values for the two parameters are $k = 0.30$ day$^{-1}$ and $D = 114$ ng mL$^{-1}$ C day$^{-1}$ respectively). Considering reported average organic aerosol (OA) concentration we assume the winter
air concentration ($C$) to be at most 2 µg C m$^{-3}$ (Guillaume et al., 2008; Holzinger et al., 2010b; Strader et al., 1999). Further taking an average sampling depth of 2 cm and a snow density of 250 mg mL$^{-1}$ we calculated a deposition velocity of 0.33 cm s$^{-1}$ according to equation (3):

$$v = \frac{D}{C \times A} \qquad (3)$$


where $D$ is the measured deposition rate, $C$ is concentration (2 µg C m$^{-3}$) and $A$ is the area that was typically sampled (combining sampling depth and snow density relates 1 mL of sample to an area of 2 cm$^2$). Assuming slightly higher (3 µg C m$^{-3}$) or lower (1 µg C m$^{-3}$) organic aerosol concentration in air, and sampling variation between 1.5-2 cm depth we calculated positive error of factor of 2 and negative
error of factor of 2$^{-1}$. Thus, deposition velocity for organic aerosols of 0.17-0.66 cm s$^{-1}$ would be



required to be consistent with the observations. However, the deposition velocities for organic aerosol were previously estimated to be $0.034 \pm 0.014$ and $0.021 \pm 0.005$ cm s$^{-1}$ for particles in 0.15–30 and 0.5–1.0 µm size ranges (Duan et al., 1988; Gallagher et al., 2002). The required deposition velocities are approximately an order of magnitude higher than the previously reported estimates even if we use the

upper limit of expected organic aerosol concentration (2 µg C m$^{-3}$). Therefore, we conclude that the dominating contribution to OM in the snow is from gas phase sVOCs. As direct measurements of bulk sVOCs do not exist, we estimated the required average loads of sVOCs in the air passing the sampling location to explain the observations. Using the deposition rate calculated from our measurements (Eq. 2), the concentration-weighted average molecular mass of measured compounds, and deposition

velocities of 1 cm s$^{-1}$ (assuming that sVOC deposition velocities are similar to the one of formic acid) (Nguyen et al., 2015), we calculated an average gas phase sVOC burden of 883 ng m$^{-3}$ of air which is equivalent to 247 ppt. Assuming slightly higher or lower deposition velocities ($\pm 0.2$ cm s$^{-1}$) yield errors of +221 and -148 ng m$^{-3}$, or +62 and -41 ppt. Our calculated value of average sVOC concentration agrees with previous estimates of 0.6 µg m$^{-3}$ (Zhao et al., 2014). Thus, our data suggest that dynamic

processes of DOM on the surface snow are dominated by deposition and re-volatilisation of gas phase sVOCs. This has important implications for our understanding of the snow surface processes. Our analysis suggests that air masses with different sVOC composition can leave different OM fingerprints in the snow (discussed in the sections below).

The $D/k$ ratio quantifies the equilibrium point (asymptote) for the model described in Eq. (2). This

represents a point at which the equilibrium is established between deposition and losses. The derived time constant for loss of about 3 days implies that 90% equilibrium is established for the total ion concentration in only 6 days. This value, however, represents an average equilibrium time for total measured DOM, and it is reasonable to assume that this equilibration timescale differs between different compounds. In particular, it is estimated to be established much faster for the gas-phase sVOCs

compared to SOA.

Similar mass balance calculations will be carried out in the following section for individual ion groups.

## 3.2 Grouping of ions with similar time evolution

In the data analysis of TD-PTR-MS spectra, we found 270 organic ions above the detection limit

present in the samples. Compounds that have the same origin (similar sources or atmospheric chemistry processes) should feature similar time evolution, if the lifetime is not so short that such a common time evolution is lost. Based on the pattern of concentration change over time (using Pearson correlation) we identified four groups of ions with similar time evolution (Fig. 3, Table A1). In the group 1, 2, 3 and 4 we assigned 25, 33, 9 and 21 ions respectively (88 ions in total 33%), and 175 ions did not fall in any of

these groups. Ions which we did not assign to any group either showed different time evolution or had concentrations close to the detection limit causing poor correlation. In average, the total concentration levels of the ions within the four groups were 30, 56, 16, 57 ng mL$^{-1}$ respectively, and 315 ng mL$^{-1}$ for ions which did not fall in any of the described groups. Specific information can be found in the supplementary data.



In the first two groups (Fig. 3a and b), among the numerous ions we identified masses that we
       tentatively attribute to pinonic acid (m/z 115.07 fragment) and levoglucosan (e.g. m/z 85.03 and 97.03
       fragments). Pinonic acid is an oxidation product of monoterpenes and the main source is expected to be
       emission from surrounding alpine conifer forests, thus group 1 ions indicate air masses that were
       originally rich in biogenic VOC, which have been processed during transport. Levoglucosan is a clear
indicator of biomass burning and the most likely source during this period is domestic wood
       combustion. Therefore, we associate group 2 ions to the anthropogenic wood combustion sources and
       their products in the complex atmospheric processing.
       The compounds that fall in group 3 show, after an initial increase in the concentration on 23/03/2017, a
       decreasing trend (Fig 3c, see also Table A1). The change in the concentration of the compounds
constituting this group may point to a one-time significant pollution event which happened between $20^{th}$
       and $23^{rd}$ of March. The total concentration of ions in this group was measured to be 34 ng mL$^{-1}$ (8.2%
       of the total organics) on 23/03/2017. This deposition event could have come from a single source,
       however higher time resolution measurements are needed to further characterize the potential source.
       As total concentration of ions in this group drops in 6 days below 20 ng mL$^{-1}$ (3.1% of the total
organics), this group is also an example how contaminated snow equilibrates with the cleaner
       atmosphere on similar timescales as we derived from the simple box model.
       As for total concentration, most of the ions and ion groups show an increase in the ion concentrations
       throughout the sampling period. Group 4 (Fig. 3d, Table A1) represents the compound group for which
       the concentration seems steadily increasing towards an equilibrium. This indicates that the simple mass
balance model may be applicable, i.e., the assumption of a (close to) constant deposition and first order
       loss rate. Therefore we applied the simple mass balance model (see 3.1.) also to the individual ions in
       group 4 to investigate whether individual organic compounds have different $k$ values. This is expected
       due to the different chemical and physical properties (such as volatility, amenability to photolysis etc.)
       as well as different nutrition adequacy for potential biodegradation. For the sum of organic ions in the
group 4 (Fig. 3d, Table A1), $k = 0.20$ day$^{-1}$. Generally, the lower $k$ value of this group compared to the
       total sVOC could be related to the fact that most of the ions here are heavier (thus, less volatile).
       However, within this group $k$ values of individual ions were found to be independent of the molecular
       weight, and also independent of the composition, i.e. O/C, H/C, OSC and nC ($R^2 < 0.12$). As the
       volatility of sVOC is expected to depend on molecular weight and functional groups (longer sVOC are
in general less volatile, unless additional functional groups are involved), this suggests that volatility
       might not play the only role in the loss processes of this group.
       A deviation in the general concentration trend of individual ions (from the expected growth, section 3.1)
       was observed on 29/03/2017, particularly in the groups 1 and 2 represented by pinonic acid and
       levoglucosan (Fig. 3a and b). Elevated levoglucosan and lower pinonic acid levels observed at $29^{th}$ are
temporally related to a change in wind direction. On the $29^{th}$, the air masses originate from the North-
       East direction, rather than North-West direction, seen for other samples (Fig 1c), so this event is
       attributed to the meteorological situation.
       Presence of such distinctive patterns of concentration change over time, ion grouping, and their relation
       with the meteorological data indicate that meteorology and deposition of sVOCs after fresh
precipitation strongly affect the organic composition in snow, which questions the most straightforward
       approach of interpreting of OM signals in terms of organic aerosol in the air.



## 3.2 Elemental composition

We further investigate the processing of OM in snow during the study period by calculating cumulative
metrics of the OM composition from the PTR-MS data, namely the elemental ratios O/C and H/C, the
number of carbon atoms per compound, nC, and the oxidative state OSC of the organic carbon, in order
to further characterize the processes behind the observed changes. The fresh snow sample (20/04/2017)
has the lowest total concentration of all measured organics, low OSC, the lowest O/C and N/C value,
and high H/C and nC value (Fig. 2), which all indicate 'fresh' OM in the air (Kroll et al., 2011), which
was captured in the snow.
An interesting signature in the metrics is observed on 29/03/2017 when the prevailing air flow regime
was interrupted (wind direction change, Fig. 1c). This sample showed the highest value of nC, the
lowest OSC, as well as elevated H/C and low O/C ratios (Fig. 2). This all indicates photochemically
younger (fresher) emissions of VOC and semi-volatiles, originating from air-masses rich in biomass
burning aerosols (Fig. 3b), which is in agreement with previous results linking low OSC and high nC to
biomass burning aerosols (Kroll et al., 2011). However, on the 29th we also observed lower total OM
concentration in the sample compared to the previous period which clearly indicates a net loss of OM.
Potential processes that could explain such loss of OM involve photolysis induced re-volatilisation, OM
runoff (e.g. snow melting), or oxidation. The photolysis induced volatilization should be higher for this
sample as the previous days (27th and 28th) had the highest global radiation values (33% higher the
average for the sampling period) and the longest sunshine duration (>12 hours) (Fig 1b and Fig. A1).
On the other hand, no significant temperature increase has been measured to support increased melting
and OM runoff. Loss by oxidation (referring to "dark" oxidation that is uncoupled from photooxidation)
is also unlikely as main process since the O/C ratio did not increase for 29th of March (Fig. 3c). Thus,
the most likely cause of the lower total OM concentration observed on march 29 is re-volatilization,
possibly enhanced by photolysis, which would indicate that the air contained a lower burden of SVOCs.
In addition, new OM material with different characteristics was deposited before that sample was
collected. Combining all metrics (Fig. 2) and meteorological data available (Fig. 1), we can conclude
that the air passing the site prior to 29/03/2017 was cleaner and photochemically younger, and
contained higher MW compounds that might have originated from anthropogenic emissions such as
biomass burning (high levels of levoglucosan, Fig. 3).

## 4 Conclusion

In this work, we analysed the concentrations of low molecular weight organic matter (20 – 500 amu) in
Alpine snow samples during a 12-day no-precipitation period, 20.03-01.04.2017. We noticed four
distinctive groups of ions with similar concentration trend over that time ($R^2 > 0.9$), suggesting common
sources, chemistry processes, or transport pathways. The largest two groups of ions came from (a)
surrounding forests (e.g. pinonic acid – associated with monoterpene oxidation) and (b) residential fires
(levoglucosan – common biomass burning marker). The snow sample taken on 29th of March showed a
change in the general concentration trend, consistent with a shift in wind direction, indicating different





air mass origin. This is also in agreement with a change in atomic ratio metrics (O/C, H/C, OSC and
nC), which also indicated that re-volatilization is the most important pathway of OM loss here,
suggesting that the advected air was cleaner during this period. Dry deposition can be approximated by
a mass balance model with a roughly constant deposition rate of $D = 206$ ng mL$^{-1}$ day$^{-1}$ and a first order
loss rate constant $k = 0.31$ day$^{-1}$. Calculated deposition velocities were inconsistent with the idea that

organic aerosols contribute the bulk of deposited OM, instead we suggest a dominant contribution of
gas-phase sVOC over the OA in the total bulk organic matter. This all indicates that, at least for this site
and location, snow-atmosphere DOM exchange processes are mostly driven by gas-phase sVOCs, for
which equilibration with air is fast. This has implications for the reconstruction of recent atmospheric
conditions by analysis of organics in the snow.

**Data availability**

Data are available via:
http://www.projects.science.uu.nl/atmosphereclimate/Data/Suplement_data_PTR-SNOW.xlsx

**Author contribution**

DM and RH designed the experiments and DM carried them out. LE provided the samples and
meteorological data. DM prepared the manuscript with contributions from all co-authors.

**Competing interests**

The authors declare that they have no conflict of interest.

**Acknowledgements**

Tis work is supported by Netherlands Earth System Science Centre (NESSC) research network and by
the Dutch NWO Earth and Life Science (ALW), project 824.14.002. We thank the operators at the
Sonnblick observatory for taking the samples.

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



**Figure 1: Meteorological data measured at Sonnblick station during the sampling: (a) temperature, (b) global radiation (c) wind
direction, (d) relative humidity. Vertical lines represent the sampling time. Note the wind direction change in the period preceding
the sampling at 29/03/2017**






**Figure 2: Total concentration of organic ions and cumulative metrics of atomic ratio distribution. (a) Total concentration in ng mL⁻¹, the line represents the fit from the simple deposition model explained in the text (Eq. 1) ;(b) H/C ratio, (c) O/C ratio, (d) N/C ratio, (e) oxidative state of carbon, (f) mean numbers of carbon. The error bars represent the standard deviation of three replicates.**





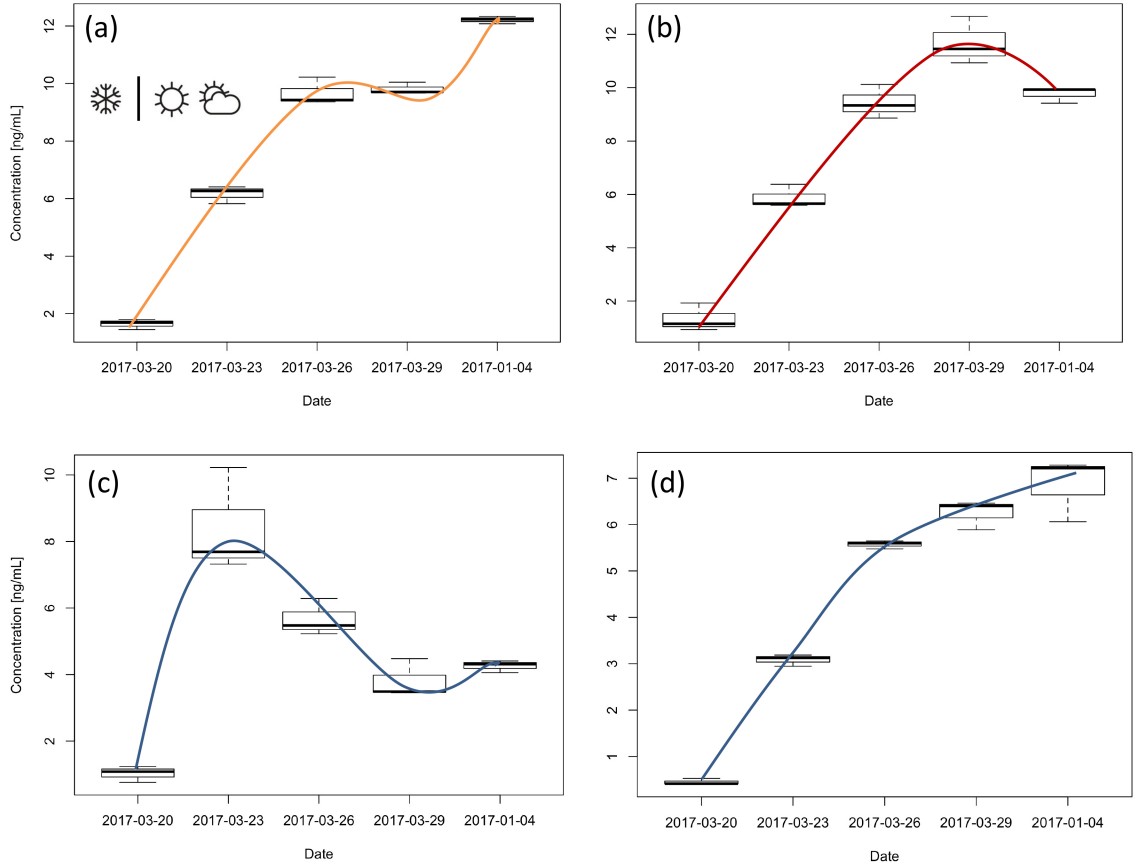

**Figure 3: Boxplots of concentration for ions representing four distinctive groups: (a) ion m/z 115.070 - pinonic acid, (b) ion m/z 85.029 - levoglucosan, (c) ions m/z 99.008 and (d) ion m/z 159.065. The lines illustrate the change in the concertation over the time that is typical for each group.**




## 5. Appendices

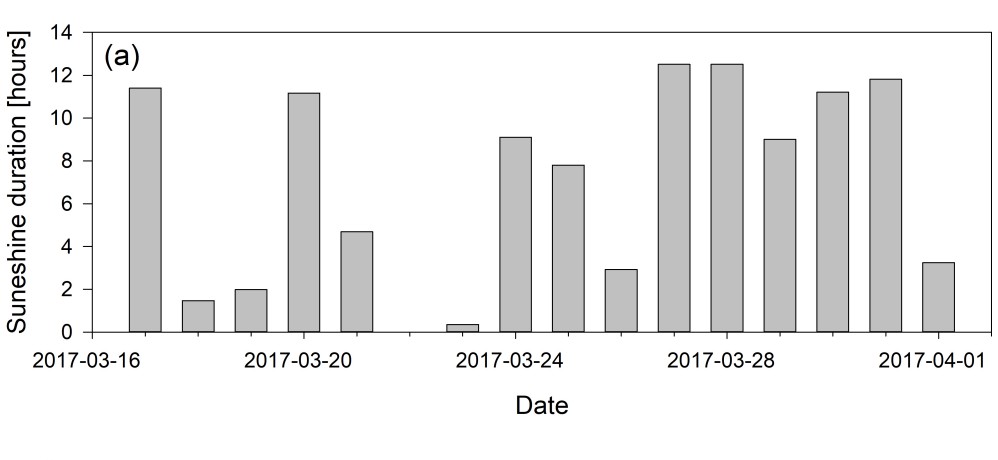

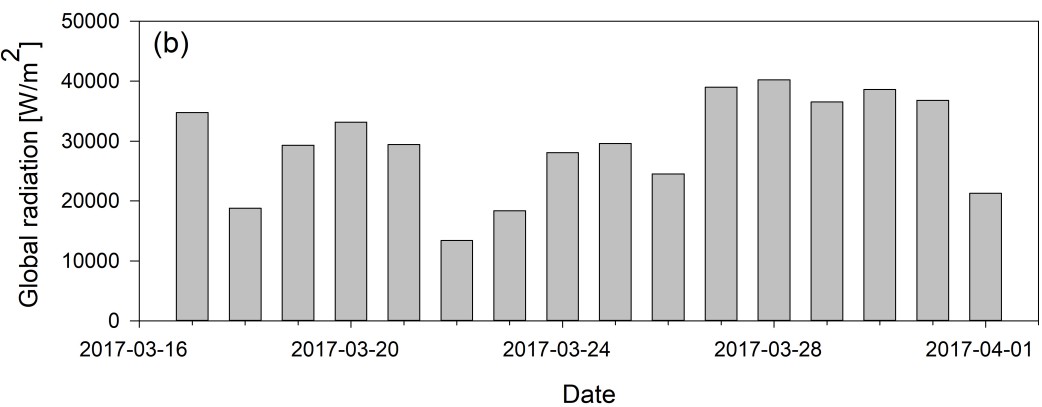

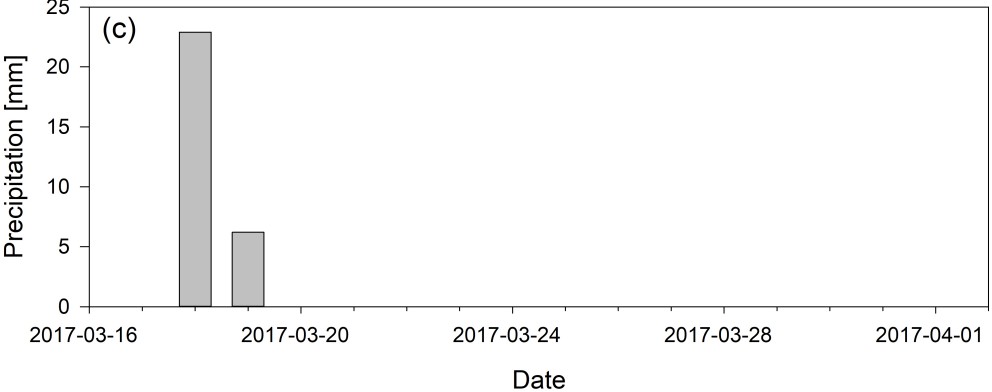

**Figure A1: Light conditions and precipitation during the sampling period. (a) Global radiation [W m⁻²] integrated for each day, (b) total daily sunshine duration in hours, (c) precipitation for the sampling period.**




**Table A1: Groups of ions as identified using linear regression model. Note that different thresholds of R² values are used to isolate the groups.**

| Pinonic acid | | Levoglucosan | | Increasing and saturating | | Decreasing | |
|---|---|---|---|---|---|---|---|
| m/z | $R^2$ | m/z | $R^2$ | m/z | $R^2$ | m/z | $R^2$ |
| 56.047 | 0.9828 | 28.017 | 0.9338 | 139.069 | 0.9953 | 80.039 | 0.7119 |
| 60.045 | 0.9823 | 31.018 | 0.9418 | 141.057 | 0.9955 | 99.008 | 1.0000 |
| 115.070 | 1.0000 | 45.033 | 0.9951 | 153.087 | 0.9980 | 113.029 | 0.9236 |
| 129.055 | 0.9810 | 53.038 | 0.9080 | 155.073 | 0.9963 | 140.040 | 0.7492 |
| 131.104 | 0.9918 | 68.052 | 0.9031 | 157.065 | 0.9988 | 163.050 | 0.9373 |
| 143.069 | 0.9919 | 69.034 | 0.9279 | 158.069 | 0.9968 | 192.057 | 0.9494 |
| 144.069 | 0.9877 | 70.033 | 0.9625 | 159.065 | 1.0000 | 193.055 | 0.9494 |
| 160.068 | 0.9832 | 70.068 | 0.9068 | 168.091 | 0.9974 | 194.050 | 0.8473 |
| 172.076 | 0.9813 | 72.046 | 0.9092 | 169.087 | 0.9970 | 357.071 | 0.9773 |
| 185.096 | 0.9871 | 73.028 | 0.9307 | 170.085 | 0.9964 | | |
| 186.091 | 0.9865 | 74.029 | 0.9523 | 171.076 | 0.9994 | | |
| 195.104 | 0.9800 | 75.043 | 0.9230 | 173.078 | 0.9958 | | |
| 197.100 | 0.9806 | 78.994 | 0.9877 | 174.078 | 0.9967 | | |
| 200.095 | 0.9801 | 82.036 | 0.9032 | 183.098 | 0.9954 | | |
| 202.092 | 0.9824 | 84.046 | 0.9548 | 184.100 | 0.9997 | | |
| 211.116 | 0.9948 | 85.029 | 0.9265 | 199.098 | 0.9955 | | |
| 213.104 | 0.9819 | 86.027 | 0.9307 | 209.127 | 0.9966 | | |
| 214.097 | 0.9803 | 86.060 | 0.9290 | 227.127 | 0.9986 | | |
| 218.086 | 0.9861 | 91.040 | 0.9349 | 235.152 | 0.9999 | | |
| 228.107 | 0.9943 | 94.038 | 0.9260 | 239.131 | 0.9951 | | |
| 230.096 | 0.9816 | 97.029 | 1.0000 | 244.095 | 0.9952 | | |
| 237.132 | 0.9828 | 97.060 | 0.9201 | | | | |
| 246.106 | 0.9845 | 100.040 | 0.9043 | | | | |
| 259.098 | 0.9821 | 108.050 | 0.9351 | | | | |
| 299.065 | 0.9882 | 109.061 | 0.9129 | | | | |
| | 0.9876 | 112.042 | 0.9003 | | | | |
| | 0.9864 | 126.057 | 0.9111 | | | | |
| | 0.9906 | 190.966 | 0.9696 | | | | |
| | 0.9946 | 226.119 | 0.9447 | | | | |
| | 0.9907 | 257.248 | 0.9258 | | | | |
| | 0.9818 | 275.263 | 0.9426 | | | | |
| | 0.9819 | 284.267 | 0.9529 | | | | |
| | | 341.339 | 0.9064 | | | | |