# Peer review of "Brief communication: Analysis of organic matter in surface snow by PTR-MS – implications for dry deposition dynamics in the Alps"

_The Cryosphere, 2018_

## Referee Comment (RC1) · Anonymous Referee #1 · 28 Nov 2018

The manuscript by Materić and colleagues deals with the deposition of organic matter/VOCs in alpine snow over a short period. They use PTR-MS, coupled with a new sample preparation method to directly analyse the dissolved organic compounds in snow. As someone who doesn't work in the field of snow/ice, I must say that I did enjoy reading the manuscript. For the most part I followed it, and the results generally back up the authors' conclusions. The paper contains a wealth of information which the authors use to make inferences about organic matter sources and weather variables.

The paper falls within the remit of The Cryosphere. Graphs are mostly well presented and easy to follow, as are tables. The purpose of the work is articulated well, the

methods seem suitable. Some additional references are probably needed.

As a non-expert I cannot comment in any great detail on some parts of the paper, particularly the mass balance approach in section 3.1. I recommend publishing the paper after various, mostly minor, comments have been addressed.

L19. VOCs needs defining on first use

L45. On L19 you say that OM originates from anthropogenic sources, biomass burning, and biogenic sources. On L45 you discuss VOC deposition. Coming to this as a non-expert, are there reasons to believe that LMW OM is an important component of the emissions from anthropogenic sources, biomass burning, and biogenic sources. Perhaps the authors could add a few references and maybe a sentence or two for the general reader.

L54. I believe that the standard for TC submissions is to use SI brochure 8 (https://www.bipm.org/utils/common/pdf/si_brochure_8_en.pdf) which would suggest either a space or no space between the 3 and 1 in 3106 m, but certainly not a comma or period.

L75. For section 2.2 there is no detail of how many or how often samples were taken. Reading into the results and it seems one (?) sample was taken every three days, but this information needs including in section 2.2.

L81. "Samples were melted." Were they melted (i.e. actively) or were they simply allowed to melt (i.e. at room temperature). Please clarify.

L81. Were filters pre-rinsed? PTFE filters can cause DOM contamination. Eg. Yoro et al, 1999, Water Research, 33, 1956-1959. https://www.sciencedirect.com/science/article/pii/S0043135498004072

L83. Sample were analysed in triplicate, but no detail is given about what happened with them. Were means taken for each ion for each set of triplicates? Additionally, whilst the authors provide a LoD, they provide no detail on the replicates. Have the

authors done analysis showing variation between replicates? This seems important.

L107. Ions m/z < 100 were excluded as the authors suggest these are mainly thermal byproducts. What evidence is there for this assumption?

L116, eq 1. What is "d"? Is it delta?

L131. 5 DPs for the R2 value is too many to be of any meaningful use, surely. The same for Table A1. My preference would be 2DPs for sufficient information.

L156 and L159. The authors make a comparison between their VOC burden (833 ng m3) and that from Zhao et al (0.6 $\mu$g m3). Please amend one of these values so the units are the same, to help the reader immediately see the comparison.

L177. The authors say they use Pearson correlation but then report R2. Technically, Pearson correlation would be R. But I wonder if a Spearman correlation might be better than Pearson. Certainly Fig. 3A would seem to follow a Spearman (i.e. monotonic increase) better than a Pearson (i.e. linear increase). Was there any reason for using Pearson?

L185 – L192. The statements concerning pinonic acid and levoglucosan origins (forests and burning) need some references added.

L193 – L201. It may not be possible, but do the authors have any suggestions for group 3? i.e. can any of the compounds be identified, thus suggesting what sort of pollution event occurred?

L208. Amenability is a slightly unusual word choice. Maybe "susceptibility" might be better.

L214 – L215. Please add a reference for "longer sVOC are in general less volatile"

L217 – L222. The authors look at changes that occurred on the 29th. Specifically, the observed increase in pinonic acid stalled. They also claim that levoglucosan was elevated, but as this was increasing anyway it seems difficult to attribute it to the same

cause? Anyway, a halt in the rise of pinonic acid would suggest (according to the earlier hypothesis involving conifer forests) that the wind has switched direction and is coming from an area less dominated by forests. Did the authors consider this? The rise in levoglucosan might suggest the wind comes from an area with more biomass burning. Again, was this considered? Fig 1 also suggests quite a pronounced humidity change on this date. What are the implications of this?

L236. Reading on I see the authors discuss biomass burning in relation to my above comment, but it still needs elaborating on.

L241. The authors suggest there is lower total OM on the 29th (fig 2A). Considering the error bars, I would say there is no difference in total OM between the 26th and the 29th.

L262. Biomass burning is mentioned previously, but here, for the first time in the paper we have mention that the source of this is residential fires. This should be mentioned earlier in the paper.

Fig 3. The labels could be increased in size, especially on the y axes, as they are difficult to read. Also, the four panels of Fig 3 and Table A1 should follow the same order. That is, Fig 3 shows pinonic (A), levoglucosan (B), decreasing trend (C), then increasing trend (D). However, in Table A1 the order is pinonic, levoglucosan, increasing, decreasing. Maybe easiest to swap columns 3 and 4 around in table A1 so the orders are the same. Also, for the legend of Fig 3, it would help the reader to note that panel C is increasing (then decreasing), and panel D is increasing (then saturating). Also, what do the weather symbols mean on panel A?

Table A1. The authors say "Note that different thresholds of R values are used to isolate the groups." This needs expanding on. Looking at Fig 3, ions that fit into panels A, B and C must all correlate with ions from other panels (they all increase to some extent). I.e. it is possible for an ion in the pinonic acid group to probably correlate with an ion in the D/increasing group. I assume that the authors are trying to say is that they used

R2 cutoffs to decide which ions went into each group. Perhaps this could be clarified.

Acknowledgements. There is a typo: "tis"

---

## Referee Comment (RC2) · Anonymous Referee #2 · 18 Dec 2018

The manuscript "Brief communication: Analysis of organic matter in surface snow by PTR-MS-implications for dry deposition dynamics in the Alps" by Materic et al., describes organic matter composition in Alpine snow samples during 12 days in spring 2017. A simple mass balance model is discussed and used to determine atmospheric deposition of VOCs on snow. A grouping method for the PTR-MS mass ions based on Pearson correlations is then used in order to highlight specific emission sources or atmospheric events that influenced the sampling site. I find the manuscript interesting, novel, and nice at reading. Specifically, it is promising the novel approach of using a state-of-the-art technique as PTR-MS, commonly used in atmospheric chemistry to monitor air samples, in the field of cryosphere. I find the manuscript suitable

to the journal and I recommend its publication after some minor comments have been addressed.

L. 85: As the approach of analysis used by the authors is quite novel it would be nice to have more details about the TD method and the PTR-MS conditions of analysis.

L. 88: How much is the percentage of recovery with the TD method for 20-500 amu? Why the maximum temperature used is 250 °C?

L. 177: Which threshold of the Pearson correlation was used to group the mass ions? Why the authors have not considered to try a more robust approach for sources apportionment as for example, the positive matrix factorization analysis?

L. 182: These numbers seem higher compared to atmospheric concentrations of a remote site. Could you include a short discussion with comparisons with reported values in literature of concentrations found in snow samples for similar compounds?

L. 210: Was any compound associated to "group 4" identified? In general, was also any other method applied simultaneously to PTR-MS analysis to cross-validate some information?

Figure 1: on 29/03/2017

Figure 3: a, b, d show a general increase. Is this due to any specific atmospheric event or driver?

Table A1: This table should be moved from the appendix to the main body of the manuscript. Here a few adjustments are needed: the text refers to Pearson coefficients but the table shows the R2. The labels of the table do not correspond to what the grouping described in the text. It is not clear which ion correlate with which. Would it be possible to the authors to re draw the table to see the correlation of each pairs of ions? Is there any of this ion identified with a compound or previously reported in literature? If yes, please mention it. How were the fragments/clusters excluded from the correlation analysis? Could you shortly discuss the possibility of having fragments

or water clusters included in the analysis?

---

## Author Comment (AC1) · 15 Jan 2019

The manuscript by Materic ÌĄ and colleagues deals with the deposition of organic matter/VOCs in alpine snow over a short period. They use PTR-MS, coupled with a new sample preparation method to directly analyse the dissolved organic compounds in snow. As someone who doesn't work in the field of snow/ice, I must say that I did enjoy reading the manuscript. For the most part I followed it, and the results generally back

up the authors' conclusions. The paper contains a wealth of information which the authors use to make inferences about organic matter sources and weather variables. The paper falls within the remit of The Cryosphere. Graphs are mostly well presented and easy to follow, as are tables. The purpose of the work is articulated well, the methods seem suitable. Some additional references are probably needed. As a non-expert I cannot comment in any great detail on some parts of the paper, particularly the mass balance approach in section 3.1. I recommend publishing the paper after various, mostly minor, comments have been addressed.

1. L19. VOCs needs defining on first use

R1. We defined the term. (L19)

2. L45. On L19 you say that OM originates from anthropogenic sources, biomass burning, and biogenic sources. On L45 you discuss VOC deposition. Coming to this as a non-expert, are there reasons to believe that LMW OM is an important component of the emissions from anthropogenic sources, biomass burning, and biogenic sources. Perhaps the authors could add a few references and maybe a sentence or two for the general reader.

R2. The references in the line below (L48-49) apply here for general information of different OM. However, due to the word (and reference number) limit, we are not able to provide a more general introduction of OM sources and fractionation.

3. L54. I believe that the standard for TC submissions is to use SI brochure 8 (https://www.bipm.org/utils/common/pdf/si_brochure_8_en.pdf) which would suggest either a space or no space between the 3 and 1 in 3106 m, but certainly not a comma or period.

R3. We corrected as required "3106 m". L55.

4. L75. For section 2.2 there is no detail of how many or how often samples were taken. Reading into the results and it seems one (?) sample was taken every three

days, but this information needs including in section 2.2.

R4. One sample was taken every three days for this analysis. We clarified it in the section 2.2, as suggested. L76.

5. L81. Samples were melted. Were they melted (i.e. actively) or were they simply allowed to melt (i.e. at room temperature). Please clarify.

R5. Samples were melted at the room temperature. We clarified this in the text L82.

6. L81. Were filters pre-rinsed? PTFE filters can cause DOM contamination. Eg. Yoro et al, 1999, Water Research, 33, 1956-1959. https://www.sciencedirect.com/science/article/pii/S0043135498004072

R6. We did not pre-rinsed. It seems that the contamination referred above does not affect PTR-MS results, as the system blanks sowed (L98-102). Also, we did not have a large volume of the samples to afford the sample lose, and we did not consider pre-rinsing with miliQ water for the dilution effect (on this small volumes).

7. L83. Sample were analysed in triplicate, but no detail is given about what happened with them. Were means taken for each ion for each set of triplicates? Additionally, whilst the authors provide a LoD, they provide no detail on the replicates. Have the authors done analysis showing variation between replicates? This seems important.

R7. The triplicates were averaged, and standard deviation is used for error bars, to express the variations between replicates (e.g. Fig 2). We added an explanation about averaging and LOD blanks in L86-88. As the Fig. 3 uses the boxplots, we clarified the meaning of the boxplot to make variation more obvious, L403.

8. L107. Ions m/z < 100 were excluded as the authors suggest these are mainly thermal byproducts. What evidence is there for this assumption?

R8. We noticed, increase level of ions m/z <100 towards the end of thermal desorption (TD). As our method loses higher volatility compounds during LPE and in the initial

phase of TD (seen as very early peaks that we exclude as we integrate after TD ramp reached 50C), we assume that these products, that show up much later, are indeed products of fragmentation due to the thermolysis.

9. L116, eq 1. What is "d"? Is it delta?

R9. We used Leibniz's notation to express the derivative.

10. L131. 5 DPs for the R2 value is too many to be of any meaningful use, surely. The same for Table A1. My preference would be 2DPs for sufficient information.

R10. For the consistency and clarity, we reduce all R2 values to 4 decimal places. L135.

11. L156 and L159. The authors make a comparison between their VOC burden (833 ng m3) and that from Zhao et al (0.6 $\mu$g m3). Please amend one of these values so the units are the same, to help the reader immediately see the comparison.

R11. Changed to 600 ng m-3. L163.

12. L177. The authors say they use Pearson correlation but then report R2. Technically, Pearson correlation would be R. But I wonder if a Spearman correlation might be better than Pearson. Certainly Fig. 3A would seem to follow a Spearman (i.e. monotonic increase) better than a Pearson (i.e. linear increase). Was there any reason for using Pearson?

R12. Change to "linear regression model", L181. For this simple correlation test, we did not consider using more advanced correlation models. (See also R3 of the Referee 2).

13. L185 – L192. The statements concerning pinonic acid and levoglucosan origins (forests and burning) need some references added.

R13. A reference added: (Salvador et al., 2016). L192.

14. L193 – L201. It may not be possible, but do the authors have any suggestions for group 3? i.e. can any of the compounds be identified, thus suggesting what sort of pollution event occurred?

R14. Not possible at the moment (see also R5 of the Referee 2).

15. L208. Amenability is a slightly unusual word choice. Maybe "susceptibility" might be better.

R15. Changed as suggested. L213.

16. L214 – L215. Please add a reference for "longer sVOC are in general less volatile"

R16. This is a general assumption in chemistry based on considering alkanes or fatty acids. Early works addressing this, e.g. (Bradley, 1954) and references within, would be outdated and thus not appropriate, considering the reference number limit of this journal.

17. L217 – L222. The authors look at changes that occurred on the 29th. Specifically, the observed increase in pinonic acid stalled. They also claim that levoglucosan was elevated, but as this was increasing anyway it seems difficult to attribute it to the same cause? Anyway, a halt in the rise of pinonic acid would suggest (according to the earlier hypothesis involving conifer forests) that the wind has switched direction and is coming from an area less dominated by forests. Did the authors consider this? The rise in levoglucosan might suggest the wind comes from an area with more biomass burning. Again, was this considered? Fig 1 also suggests quite a pronounced humidity change on this date. What are the implications of this?

R17. We considered the "rise in levoglucosan might suggest the wind comes from an area with more biomass burning." We further elaborate this in L227. Humidity might be related to the change in the height of the boundary layer, but we chose not to address this at the time. More data is needed for further conclusions regarding this.

18. L236. Reading on I see the authors discuss biomass burning in relation to my

above comment, but it still needs elaborating on.

R18. See the comment R17.

19. L241. The authors suggest there is lower total OM on the 29th (fig 2A). Considering the error bars, I would say there is no difference in total OM between the 26th and the 29th.

R19. To be more accurate, and to agree with the discussion afterwards, we corrected it to "...we also observed lower average total OM concentration...". L247.

20. L262. Biomass burning is mentioned previously, but here, for the first time in the paper we have mention that the source of this is residential fires. This should be mentioned earlier in the paper.

R20. It has been introduced in L194 when first time used "biomass burning".

21. "Fig 3. The labels could be increased in size, especially on the y axes, as they are difficult to read." Also, the four panels of Fig 3 and Table A1 should follow the same order. That is, Fig 3 shows pinonic (A), levoglucosan (B), decreasing trend (C), then increasing trend (D). However, in Table A1 the order is pinonic, levoglucosan, increasing, decreasing. Maybe easiest to swap columns 3 and 4 around in table A1 so the orders are the same. Also, for the legend of Fig 3, it would help the reader to note that panel C is increasing (then decreasing), and panel D is increasing (then saturating). Also, what do the weather symbols mean on panel A?

R21. Fig 3 labels increased in size. Columns in the table replaced as suggested. The lines added in the Fig 3 a-d should be sufficient to describe the trend: "The lines illustrate the change in the concertation over the time that is typical for each group" L410. We further explained the weather symbols in the figure caption (L411-413).

22. Table A1. The authors say "Note that different thresholds of R values are used to isolate the groups." This needs expanding on. Looking at Fig 3, ions that fit into panels A, B and C must all correlate with ions from other panels (they all increase to some

extent). I.e. it is possible for an ion in the pinonic acid group to probably correlate with an ion in the D/increasing group. I assume that the authors are trying to say is that they used R2 cutoffs to decide which ions went into each group. Perhaps this could be clarified.

R22. We further clarified the table caption as suggested - L415. R2 cutoffs also added for clarity.

23. Acknowledgements. There is a typo: "tis"

R23. We corrected this.

Anonymous Referee #2

The manuscript "Brief communication: Analysis of organic matter in surface snow by PTR-MS-implications for dry deposition dynamics in the Alps" by Materic et al., describes organic matter composition in Alpine snow samples during 12 days in spring 2017. A simple mass balance model is discussed and used to determine atmospheric deposition of VOCs on snow. A grouping method for the PTR-MS mass ions based on Pearson correlations is then used in order to highlight specific emission sources or atmospheric events that influenced the sampling site. I find the manuscript interesting, novel, and nice at reading. Specifically, it is promising the novel approach of using a state-of-the-art technique as PTR-MS, commonly used in atmospheric chemistry to monitor air samples, in the field of cryosphere. I find the manuscript suitable to the journal and I recommend its publication after some minor comments have been addressed.

1. "L. 85: As the approach of analysis used by the authors is quite novel it would be nice to have more details about the TD method and the PTR-MS conditions of analysis."

R1: Unfortunately, space is limited in this brief communication format so include more on the method. However, our method paper (reference provided in the text) is published

as an open-source so the information on the method is easily accessible. PTR-MS conditions are additionally explained in the text L88-89.

2. "L. 88: How much is the percentage of recovery with the TD method for 20-500 amu? Why the maximum temperature used is 250 âŮęC? "

R2. The exact percentage of recovery for this samples are unfortunately not available as we did not measure dissolved organic matter concentrations. From our previous works, we had the mean recovery of 0.6% desorbing at maximum 240 °C (5-minute protocol) (Materic 2017), and later 5.5% at 250 °C (8-minute protocol) (Peacock 2018). We followed our experience and used 250 °C and 8-minute protocol, so we expect ∼5% recovery.

3. L. 177: Which threshold of the Pearson correlation was used to group the mass ions? Why the authors have not considered to try a more robust approach for sources apportionment as for example, the positive matrix factorization analysis?

R3. In this pilot, for the ion correlation we used the linear regression for its simplicity (thresholds: group 1 and 2 R2>0.98 saturating > 0.995, decreasing R2>0.70. PMF). PMF is however planned for the experiments which involve more frequent sampling rate.

4. L. 182: These numbers seem higher compared to atmospheric concentrations of a remote site. Could you include a short discussion with comparisons with reported values in literature of concentrations found in snow samples for similar compounds?

R4. Our concentration seems realistic compared to the literature values for the same location (Sonnblick Observatory, AU), although different methods have been used. E.g. in (Gröllert et al., 1997), they measured aliphatic hydrocarbons at 14 ng/mL (ïĄ∎g/L reported), aliphatic alcohols 18 ng/mL and fatty acids 43 ng/mL. We added the discussion in L188-189.

5. "L. 210: Was any compound associated to "group 4" identified? In general, was also

any other method applied simultaneously to PTR-MS analysis to cross-validate some information? "

R5. We did not perform GC-MS or similar for cross-validation at this stage. So, we are not certain for the group 4 ion identification.

6. Figure 1: on 29/03/2017

R6. Corrected to the same format L401.

7. Figure 3: a, b, d show a general increase. Is this due to any specific atmospheric event or driver?

R7. As the snow is exposed to the air (which has certain levels of OM), the dry deposition it is expected to increase toward the equilibrium (see L115 and the following lines).

8. Table A1: This table should be moved from the appendix to the main body of the manuscript. Here a few adjustments are needed: the text refers to Pearson coefficients but the table shows the R2. The labels of the table do not correspond to what the grouping described in the text. It is not clear which ion correlate with which. Would it be possible to the authors to re draw the table to see the correlation of each pairs of ions? Is there any of this ion identified with a compound or previously reported in literature? If yes, please mention it.

R8. Instead of "Pearson..." we used "linear regression model". The caption of the table improved. Table columns adjusted. Correlation of each pair would be complicated to present in one table, instead, we gave correlations against a "typical" ion of the group, highlighted bold. Using the limited literature sources for TD-PTR-MS, no further identification of the ions is possible at this stage (see also R5). The limited number of figures and tables in the brief communication format prevents us to move the table to the main text.

9. How were the fragments/clusters excluded from the correlation analysis? Could

you shortly discuss the possibility of having fragments or water clusters included in the analysis?

R9. We reduced the effect of fragments by excluding low molecular mass compounds (comment R8 referee 1.). Water clusters are not addressed here as we measure at reasonably high E/N (122 Td). More detailed discussion on the fragments/clusters impact on the analysis is planned for a different experiment where we measured using different E/N, in which case this could be properly addressed.
* * *

---

## Author Response (AR2)

Comments to the Author:

Dear Authors thank you for the revision. The manuscript is publishable with the corrections made, except that the data statement and availability must be improved (see also https://www.the-cryosphere.net/about/data_policy.html ). Before final acceptance, the data must be put in a permanent data repository, sufficient metadata (location, detailed sampling pr added, and the complete dataset (i.e. non-averaged values) must be presented.

**Respond:**

**We submitted our data to a public repository. We edited the metadata accordingly, and we changed the link in the manuscript L283 adding the DOI: https://doi.org/10.24416/uu01-6ly8gt**

[revised manuscript text omitted]